# Research of Low-cost Air Quality Monitoring Models with Different Machine Learning Algorithms

Gang Wang[1, 2, 3], Chunlai Yu[1, 3], Kai Guo[2], Haisong Guo[1, 3], Yibo Wang[2]

[1]Huanghe Science and Technology College, Zhengzhou 450063, China

[2]Hanwei Electronics Group Corporation, Zhengzhou 450001, China

[3]Zhengzhou Key Laboratory of Intelligent Measurement Techniques and Applications, Zhengzhou, 450063, China

*Correspondence to*: Gang Wang (wywanggang163@163.com)

**Abstract.** To improve the performance of the calibration model for the air quality monitoring, a low-cost multi-parameter air quality monitoring system (LCS) based on different machine learning algorithms is proposed. The LCS can measure particulate matter ($PM_{2.5}$ and $PM_{10}$) and gas pollutants ($SO_2$, $NO_2$, CO and $O_3$) simultaneously. The multi-input multi-output (MIMO) prediction model is developed based on the original signals of the sensors, ambient temperature ($T$) and relative humidity ($RH$), and the measurements of the reference instrumentations. The performance of the different algorithms (RF, MLR, KNN, BP, GA-BP) with the parameters such as determination coefficient $R^2$, Root Mean Square Error (RMSE), mean square error (MSE) and mean absolute error (MAE) are compared and discussed. Using these methods, the $R^2$ of the algorithms (RF, MLR, KNN, BP, GA-BP) for the PM is in the range 0.68 - 0.99; the RMSE values of $PM_{2.5}$ and $PM_{10}$ are within 2.36 - 18.68 $\mu gm^{-3}$ and 4.55 – 45.05 $\mu gm^{-3}$, respectively; the MAE values of $PM_{2.5}$ and $PM_{10}$ are within 1.44 - 12.80 $\mu gm^{-3}$ and 3.21- 23.20 $\mu gm^{-3}$, respectively. The $R^2$ of the algorithms (RF, MLR, KNN, BP, GA-BP) for the gas pollutants ($O_3$, CO and $NO_2$) is within 0.70 - 0.99; the RMSE values for these pollutants are 4.05 - 17.79 $\mu gm^{-3}$, 0.02 - 0.18 $mgm^{-3}$, 2.88 - 14.54 $\mu gm^{-3}$, respectively; the MAE values for these pollutants are 2.76 - 13.46 $\mu gm^{-3}$, 0.02 - 0.19 $mgm^{-3}$, 1.84 - 11.08 $\mu gm^{-3}$, respectively. The $R^2$ of the algorithms (RF, KNN, BP, GA-BP, except for MLR) for $SO_2$ is within 0.27 - 0.97, the RMSE value is in the range 0.64 -5.37 $\mu gm^{-3}$, and the MAE value is in the range 0.39 - 4.24 $\mu gm^{-3}$. These measurements are consistent with the national environmental protection standard requirement of China, and the LCS based on the machine learning algorithms can be used to predict the concentrations of PM and gas pollution.

## 1 Introduction

The development along with increased population and urbanization brings disadvantages, such as decreasing air quality and impact on public and individual health (Khreis et al., 2022; Manisalidis et al., 2020; Singh et al., 2021). Among the atmospheric pollutants, the primary pollutant is fine particulate matter, which affects the respiratory system and cardiac activity of humans. The secondary pollutants are $SO_2$, CO, $NO_x$, and $O_3$, which also induce disease or chronic poisoning. To improve the understanding of air pollution exposure and predict future air quality trends(Zimmerman et al., 2018), air quality assessment and forecasting are the essentials. The conventional air quality monitoring instrumentations are high cost, which has limited the spatial coverage of the monitoring stations(Zimmerman et al., 2018). The development and applications of the low-cost commercially available sensor-based air quality monitoring system (LCS) would considerably reduce both installation and maintenance costs (Spinelle et al., 2017). The larger spatial density of the air quality grid monitoring network becomes possible, which would play an important role in monitoring pollution trend, locating of pollution source, supporting environmental management(Zhao et al., 2019) and support better epidemiological models(Khreis et al., 2022; Zimmerman et al., 2018). These demands promote the LCS growing gradually(Cui et al., 2021; Wang et al., 2016).

The LCS typically utilizes the electrochemical or light scattering sensors for gas-phase or particulate pollutants measurement, such as sulfur dioxide ($SO_2$), nitrogen oxide ($NO_2$), carbon monoxide (CO), ozone ($O_3$) and particulate matters (PM). These electrochemical sensors have intrinsic problems, such as temperature or humidity impacts, and gaseous cross-sensitivities (Spinelle et al., 2015, 2017; Wan et al., 2016; Zimmerman et al., 2018). For example, limited by the poor selection performance, the $NO_2$ electrochemical sensor also undergo redox reactions in the presence of $O_3$ gaseous pollutants. The diffusion coefficient of the electrochemical sensor can be affected by temperature and relative humidity(Hitchman et al., 1997; Masson et al., 2015). The reagent of the electrochemical sensor is consumed over time, which affects the stability of the sensor. These features of the sensors have historically been poorly addressed by laboratory calibrations, limiting the utility for air quality monitoring (Zimmerman et al., 2018).

The de-convolving of cross-sensitivity effect and stability on sensor performance is complex(Zimmerman et al., 2018). The linear or multivariate linear calibration models (Alexopoulos, 2010; Khreis et al., 2022; Zoest et al., 2019) have been developed. However the performance is poor on ambient data(Khreis et al., 2022). The accurate and precise calibration models for the low cost sensors are particularly critical to the success of dense sensor networks, as poor signal to noise ratios and cross-sensitivities hamper their ability to distinguish the pollutant concentrations. There has been increasing interest in multifarious algorithms for low-cost sensor calibration, and lots of studies using multi-input multi-output models(Alexopoulos, 2010) and neural networks(Spinelle et al., 2015) have been published. The artificial neural network (ANN) calibration model has the intelligence to process nonlinear data (Amuthadevi et al., 2021; Janabi et al., 2021), which has been used in calibration models for measuring ozone or nitrogen oxide(Esposito et al., 2016; Spinelle et al., 2015). For example, the ANN calibration model was used to calibrate $O_3$ and the uncertainty could meet the European data quality objectives; however, meeting these objectives for $NO_2$ remained a challenge(Spinelle et al., 2015). Dynamic neural network calibrations of $NO_2$ sensors were demonstrated with the mean absolute error less than 2 ppb; however, the performance for $O_3$ was not same(Esposito et al., 2016). High-dimensional multi-response model was used to calibrate CO, NO, NO2, and O3, with the 5 min average RMSE values of 39.2, 4.52, 4.56, and 9.71, respectively(Cross et al., 2017). Random-forest-based machine learning algorithm was used to improve the calibration strategies of low-cost sensors, with the mean absolute error values 38 ppb for CO, 10 ppm for $CO_2$, 3.5 ppb for $NO_2$, and 3.4 ppb for $O_3$, respectively(Zimmerman et al., 2018). Furthermore, multiple linear regression(Ionascu et al., 2021) based temperature and humidity correction and ANN based calibration shown the potential for significant further improvement for leave one out cross validation(Ali et al., 2021). With the 16 days process, the combined supervision calibration model was used to improve the $R^2$ of $SO_2$, $NO_2$ and $O_3$ by 75.8%, 38.6% and 4.7% to 0.58, 0.61, and 0.90, respectively(Cui et al., 2021). An integrated genetic programming dynamic neural network model was used to accurately estimate the carbon monoxide and nitrogen dioxide pollutant concentrations from the multi-sensor measurement data(Davut et al., 2022). A predictive model using multilayer perceptron, support vector regression, and linear regression was developed to analyze the $CO_2$ and particulate matter of in-vehicle, with the $R^2$ of 0.9981(Chew et al., 2021). The CNN, LSTM-CNN, and CNN-LSTM models were used to improve the prediction performance of the ozone by 3.58%, 1.68%, and 3.37%, respectively(Reza et al., 2023). However, these calibrations have only been tested utilizing fewer models with a short measurement period and small number of sensor matrix, each containing one sensor per pollutant (Cross et al., 2017; Esposito et al., 2016; Spinelle et al., 2015), not have been utilized to evaluate and predict the concentration values of multi pollutants simultaneously, such as $PM_{2.5}$, $PM_{10}$, $SO_2$, $NO_2$, CO and $O_3$.

The random-forest (RF)(Breiman, 2001; Liu et al., 2012), multivariate linear regression (MLR)(Alexopoulos, 2010), K Nearest Neighbor (KNN)(Zhao et al., 2021), BP neural network(Xu et al., 2021), and genetic-algorithm-back-propagation neural (GA-BP) network(Ning et al., 2019; Wang et al., 2019) are five commonly used machine learning algorithms with different characteristics. With the strong nonlinear mapping ability and adaptive ability, the RF, BP, and GA-BP are suitable for processing complex, high-

dimensional, and nonlinear data with high prediction accuracy, such as the air quality monitoring. With the purpose to quantify the
degree of influence of the independent variable, the MLR is suitable for evaluating the influence of multiple independent variables
on the dependent variable, such as the cross-sensitivity effect between different factors. The KNN is also a widely common
algorithm to compare with RF, BP, GA-BP and MLR.
In this work, the LCS is developed to measure $PM_{2.5}$, $PM_{10}$, $SO_2$, $NO_2$, CO and $O_3$ simultaneously, and the performances of the
calibration strategies based on the five machine learning algorithms are contrasted. Taking the original electronic signals of the
sensors as input and measurements obtained by the reference instrumentations as output, five calibration strategies are applied and
contrasted. The measurement is implemented in the real-world conditions almost a 12-month period (1 March 2021 and 28
February 2022) spanning multiple seasons and a wide range of meteorological conditions to ensure calibration model robustness.
The performance of the different algorithms with the parameters, such as determination coefficient ($R^2$), root mean square error
(RMSE) (Janabi et al., 2021), mean square error (MSE) and mean absolute error (MAE), are compared and discussed. The rest of
this paper is organized as follows. The measurement setup is described in section 2. The principles of the calibration strategies are
presented in section 3. The results and discussion are shown in section 4. The conclusion and discussion are drawn in section 5.
**2 Measurement setup**
This section describes the measurement site and data collection, schematic block of the LCS, and the reference instrumentation.
The low-cost here is defined as below 150 dollars per pollutant, commercial availability and low maintenance. The sensors typically
utilize electro-chemical signal and scattering light intensity for gas-phase pollutants ($SO_2$, $NO_2$, CO and $O_3$) and particle pollutants
($PM_{2.5}$, $PM_{10}$) measurement.
**2.1 Measurement site and data collection**

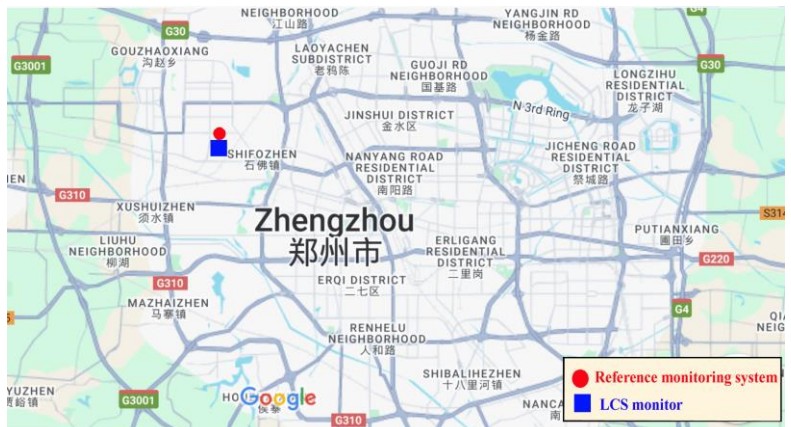

**Figure 1 Location of the air quality monitoring station during the measurement period**

Measurements for gas-phase pollutants and particle pollutants were made continuously between 1 March 2021 and 28 February
2022, which were used as the start and end dates for the analyses. The location, shown in Figure 1, was 30 Yaochang Streat,
Zhongyuan District, Zhengzhou City, Henan Province of China. There was an independent reference monitoring system for $PM_{2.5}$,
$PM_{10}$, CO, $SO_2$, $NO_2$ and $O_3$ measurement. The LCS was mounted at a consistent height with the reference monitoring system.
The time taken for one set of data collection was one minute and repeated 4 times. The outlier of the 4 sets of data was eliminated
by using the Dixon principle. The remained data was used to get the mean values for each experiment. The values of the LCS and

1    reference instruments were separately logged to the server with the interval of 5 minutes. During the measurement period, the

2    ranges of the ambient temperature and relative humidity separately were minus 5°C to plus 50°C and 10% to 98%.

## 2.2   Schematic block of LCS

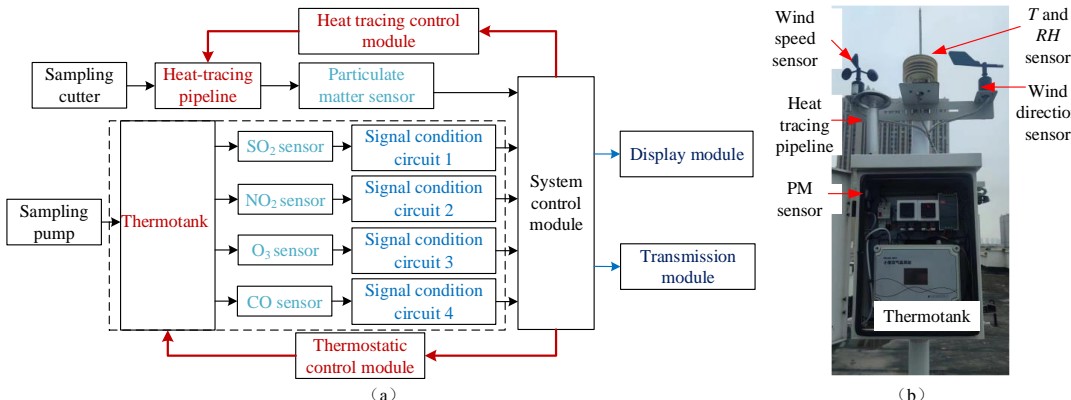

(a)                 (b)

**Figure 2. Schematic block and site photo of the LCS. The left panel (a) is the schematic block of the LCS. The system control module can ensure the temperature stability of the heat tracing pipeline and thermo-tank through the heat tracing control module and thermostatic control module, respectively. The sampling cutter is used to filter particles larger than 10 µm. The sampling pump is utilized to deliver ambient air to the surface of the sensors. The right panel (b) is the site photo of the LCS.**

In this study, the LCS is developed by Hanwei Electronics Group Corporation, and its schematic block diagram is shown in Figure 2.
The LCS uses the commercially available particulate matter sensor (PM3006, Cubic sensor and Instrument Co., China) and
electrochemical $SO_2$, $NO_2$, $O_3$, CO sensors (B4, Alphasense, UK), respectively. The particulate matter sensor device is the laser
diode (LD) based particle sensor, using a spectrophotometer to measure the particle scattering light intensity. The PM sensor device
(PM3006) can measure size dependent $PM_{2.5}$ and $PM_{10}$ concentration of the particles in the size range of 0.3 to 10 µm. The gas
pollution ($SO_2 \backslash NO_2 \backslash O_3 \backslash CO$) sensor used are with 4 electrodes (i.e. reference, worker, counter and auxiliary electrodes), where the
auxiliary electrode is not exposed to the target analyte to account for changes in the sensor baseline signal under different
meteorological conditions(Mead et al., 2013).
The electrochemical sensor outputs are measured using electronic circuitry designed by Hanwei and optimized for signal stability.
The circuitry is developed with custom electronics to drive the device, multiple stages of filtering circuitry for specific noise
signatures, and an analog-to-digital converter for measurement of the conditioned signal.
Due to the redox reaction on the anode and the cathode of the electrochemical sensor, the movement of charge between the
electrodes produces a current proportional to the analyte reaction rate, which can be used to determine the analyte
concentration(Mead et al., 2013) and the sensor whether working effectively.
Before installed into the LCS, calibrated with the different models and used in real-world conditions, the performance of the sensors
should be checked in laboratory. The linearity of the gas sensors was tested under steadily increased concentration, which was
from 0 - 5 mgm$^{-3}$ for CO sensor, 0 - 0.2 mgm$^{-3}$ for $NO_2$, 0 - 1.1 mgm$^{-3}$ for $O_3$ and 0 - 1.4 mgm$^{-3}$ for $SO_2$ with five more test points,
shown in Figure 3. Since the units of outputs of the reference instruments and the sensors were different, the slope was not expected
to be 1(Cui et al., 2021). As shown in Figure 3, the $R^2$ for the gas sensors are more than 0.93, which indicated that these gas sensors
have good linear responses before calibration, and verified the sensor working properly and effectively and could be applied to the
LCS.

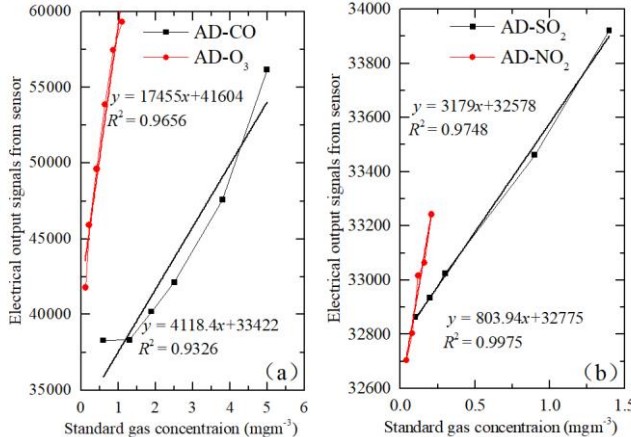

**Figure 3. Linearity of gas sensors before calibration. Electrical output signals versus single standard gas concentration is tested in**
**laboratory condition. The left panel (a) and right panel (b) represent the proportional relations between CO and O₃ sensors, SO₂ and**
**NO₂ sensors, respectively. The duration of each measurement is about 30 minutes.**
However, even with an auxiliary electrode, electrochemical sensors may insufficiently account for the impacts of temperature and
relative humidity. With the standard gases through the test chamber and the concentrations stabilized at 27 ppb for SO₂, 3.9 ppb
for NO₂, 13 ppb for O₃, and 1.22 ppm for CO, the output voltages of the four types of gaseous sensors are nonlinearly fluctuated
with the linearly increasing temperature and the relative humidity (RH)(Cui et al., 2021). With the purpose to eliminate the
influence of the external environment on the sensor as much as possible, the particles flow through a sampling cutter and heat-
tracing pipeline to the particulate matter sensor, and the gaseous pollutants are pumped to the electrochemical sensors, which are
secured in a thermo-tank. The temperature values of the heat-tracing pipeline and thermo-tank can be maintained at 60 ℃±2 ℃ to
reduce the influence of relative humidity and  25 ℃ ±2 ℃(Wei et al., 2018) to keep the sensor operating at a stable temperature,
respectively.
The measurement results of particulate matter sensor and gas pollution sensors, transmitted to the system control module through
the data buses, are directly displayed on the local display module and wirelessly transmitted to the corresponding online server
through the transmission module. As the uni-variate linear models does not in-corporate any cross-sensitivities to other pollutants
or any nonlinearities in the response, we attempt to using the sensor electronic results as the input and the reference measurements
as the output, to build multi-dimensional multi-response prediction models to de-convolve the effects of cross-sensitivity and
stability on sensor performance utilizing MLR, RF, KNN, BP and GA-BP calibration models.
**2.3  Reference instrumentation**
In order to reduce the adsorption effect on particle matter and gaseous pollutants, the reference measurements are made on ambient
air continuously drawn through Teflon fluorinated ethylene propylene (FEP)(Wei et al., 2018) tubing with a six-port stainless steel
manifold for flow distribution to the gas analyzers and particulate monitors(Mead et al., 2013). It should be pointed out that the
LCS was mounted at a consistent height with the reference monitoring system during the measurement period.
The reference ambient particulate monitor 5014*i*, which uses beta attenuation of the ambient particulate deposited onto a filter tape,
is applied to measure the mass concentration of suspended and refined particulates. The reference NO-NO₂-NOₓ monitor 42*i,* using
the linear proportional of the chemi-luminescence reaction of NO and O₃ after NO₂ transformed into NO, is utilized to measure the
NO₂ concentration. The SO₂ reference analyzer is 43*i* using the ultraviolet light (which is emitted as the excited SO₂ molecules
decay to lower energy states) intensity proportional to the SO₂ concentration. The CO reference monitor is 48*i* utilizing the principle
that CO absorbs infrared radiation at a wavelength of 4.6 μm and the infrared absorption can be transformed to be proportional to
the CO concentration. The 49$i$ O$_3$ analyzer operates on the principle that O$_3$ molecules absorb UV light at a wavelength of 254 nm,
and the absorption intensity of the UV light is directly related to the ozone concentration. All these reference monitors are produced
by Thermo Fisher Scientific Inc. The time interval for all reference measurements is 5 minutes. According to the technical
specifications for operation and quality control of ambient air quality continuous automated monitoring system for SO$_2$, NO$_2$, O$_3$
and CO of China(China, 2018), and the technical guide for automatic monitoring by beta ray method for particulate matter in
ambient air (PM$_{10}$ and PM$_{2.5}$) (China, 2020), the reference gas and particulate analyzers are checked and calibrated weekly and
monthly, respectively.
**3 Principles**
This section describes the principles of the calibration methods, such as MLR, BP, GA-BP, KNN and RF, and the metrics for
performance evaluation. The calibration models are constructed with the sensors (i.e., PM$_{2.5}$, PM$_{10}$, CO, SO$_2$, NO$_2$ and O$_3$ sensors.)
electronic results as the input and the reference measurements as the output.
**3.1  Calibration methods**
**3.1.1 Multiple linear regression model**
After the data collected by the LCS, the raw data should be preprocessed. The PM3006 particulate matter sensor can output six
kinds of particle range (i.e., >0.3μm, >0.5μm, >1.0μm, >2.5μm, >5.0μm and >10μm, respectively). By subtracting the six particle
range values in turn, the individual particle counters are obtained, and expressed as $x_{0.5}$, $x_{1.0}$, $x_{2.5}$, $x_{5.0}$ and $x_{10.0}$, listed in Table 1, the
measured particle number concentration is converted to PM mass concentrations in the PM$_{2.5}$ and PM$_{10}$ size fractions.
**Table 1. Size range of the particulate matter sensor. The sensor can measure particles with the size range of 0.3~0.5 μm, 0.5~1.0 μm,**
**1.0~2.5 μm, 2.5~5.0 μm and 5.0~10 μm, simultaneously. The corresponding particle counters are expressed as $x_{0.5}$, $x_{1.0}$, $x_{2.5}$, $x_{5.0}$ and $x_{10.0}$,**
**respectively.**

| Range (μm) | 0.3~0.5 | 0.5~1.0 | 1.0~2.5 | 2.5~5.0 | 5.0~10.0 |
|---|---|---|---|---|---|
| Particle counter | $x_{0.5}$ | $x_{1.0}$ | $x_{2.5}$ | $x_{5.0}$ | $x_{10.0}$ |

Taking the particle counters, listed in Table 1, as input and the concentrations $Y_{pm2.5}$ and $Y_{pm10}$ of PM$_{2.5}$ and PM$_{10}$ measured by
5014$i$ as output, the multivariate linear regression (MLR) models(Alexopoulos, 2010; Zoest et al., 2019) is built. Due to the
previously established influence of  ambient temperature ($T$) and relative humidity ($RH$) on sensor response(Masson et al., 2015;
Wan et al., 2016), the particle counter terms are pretreated and individual from each other. The multi-input one-response
preprocessing and prediction models can be written as Eq. (1) to obtain the concentrations $Y_{pm2.5}$.
$Y_{pm2.5}=w_{1\_pm2.5} \cdot x_{0.5} + w_{2\_pm2.5} \cdot x_{1.0} + w_{3\_pm2.5} \cdot x_{2.5}+w_{4\_pm2.5} \cdot T+w_{5\_pm2.5} \cdot RH + b_{pm2.5}$,                    (1)
Where $W_{pm2.5}= [w_{1\_pm2.5}, w_{2\_pm2.5}, w_{3\_pm2.5}, w_{4\_pm2.5}, w_{5\_pm2.5}]$ is the corresponding weight coefficients; the $X_{pm2.5} = [x_{0.5}, x_{1.0}, x_{2.5}, T,$
$RH]$ represents the individual particle counters, the temperature sensor and humidity sensor; the $b_{pm2.5}$ is the intercept values of the
model.
To obtain the concentration $Y_{pm10}$, the multi-input one-response preprocessing and prediction models can be written as Eq. (2).
$Y_{pm10}=w_{1\_pm10} \cdot x_{0.5} + w_{2\_pm10} \cdot x_{1.0} + w_{3\_pm10} \cdot x_{2.5}+w_{4\_pm10} \cdot x_{5.0}+w_{5\_pm10} \cdot x_{10.0}+w_{6\_pm10} \cdot T+w_{7\_pm10} \cdot RH + b_{pm10}$,       (2)
Where $W_{pm10} = [w_{1\_pm10}, w_{2\_pm10}, w_{3\_pm10}, w_{4\_pm10}, w_{5\_pm10}, w_{6\_pm10}, w_{7\_pm10}]$ is the corresponding weight coefficients; the $X_{pm10} =$
$[x_{0.5}, x_{1.0}, x_{2.5}, x_{5.0}, x_{10.0}, T, RH]$ represents the individual particle counters, the temperature sensor and humidity sensor; the $b_{pm10}$ is
the intercept values of the model.
Due to the poor selection performance and cross interference of the electro-chemical sensors response, the output values from each
sensor using net sensor response to the target analyte, such as $O_3$, $NO_2$, $SO_2$, concentration measured by the inference monitor are
used to build the MLR model. The CO gaseous pollution is also one of the criteria pollutants, which is must to be measured in
China. Thus, the multi-dimensional multi-response preprocessing and prediction model for the 4 gas pollutions, $T$ and $RH$ can be
written as Eq. (3).

$$\begin{cases} Y_{SO_2}= w_{11}\cdot x_{SO_2}+w_{12}\cdot x_{NO_2}+w_{13}\cdot x_{CO}+w_{14}\cdot x_{O_3}+w_{15}\cdot T+w_{16}\cdot RH+b_{SO_2} \\ Y_{NO_2}= w_{21}\cdot x_{SO_2}+w_{22}\cdot x_{NO_2}+w_{23}\cdot x_{CO}+w_{24}\cdot x_{O_3}+w_{25}\cdot T+w_{26}\cdot RH+b_{NO_2} \\ Y_{CO}= w_{31}\cdot x_{SO_2}+w_{32}\cdot x_{NO_2}+w_{33}\cdot x_{CO}+w_{34}\cdot x_{O_3}+w_{35}\cdot T+w_{36}\cdot RH+b_{CO} \\ Y_{O_3}= w_{41}\cdot x_{SO_2}+w_{42}\cdot x_{NO_2}+w_{43}\cdot x_{CO}+w_{44}\cdot x_{O_3}+w_{45}\cdot T+w_{46}\cdot RH+b_{O_3} \end{cases},$$
(3)

The equation (3) can be simplified as,
$Y_{[SO_2, NO_2, CO, O_3]}=W_{gas}\cdot X_{gas} + B_{gas}$,
(4)

Where $W_{gas}=\begin{bmatrix} w_{11} & w_{12} & w_{13} & w_{14} & w_{15} & w_{16} \\ w_{21} & w_{22} & w_{23} & w_{24} & w_{25} & w_{26} \\ w_{31} & w_{32} & w_{33} & w_{34} & w_{35} & w_{36} \\ w_{41} & w_{42} & w_{43} & w_{44} & w_{45} & w_{46} \end{bmatrix}$ is the corresponding weight coefficient; the $X_{gas} =[x_{SO_2}, x_{NO_2}, x_{CO}, x_{O_3}, T,$
$RH]$ is the convertor output values of the sensors through the electronic circuitries; the $B_{gas} = [b_{SO_2}, b_{NO_2}, b_{CO}, b_{O_3}]$ is the
intercept value of the model.
Hereto, the multi MLR models for the gas sensor and PM sensor are separately developed. The training data is used to calculate
the model regression coefficient and intercept values, and the withheld testing data is utilized to evaluate the performance of the
model performance.
**3.1.2 BP neural network model**
The BP neural network algorithm is one of the most widely used ANN models. It is a multi-layer feed-forward network trained
through an error back propagation algorithm by constantly adjusting the weight and intercept of the network. The feed-forward
topological structure of the BP neural network model, shown in Figure 4, includes the input layer, hidden layer and output layer.
With the purpose to avoid the numerical problems caused by the extreme values of polarization, eliminate the misleading effects
for feature extraction and obtain the accurate estimation of pollutant concentrations(Janabi et al., 2021), the collected input sensor
date $X_I$ and output date $Y_O$ should be respectively normalized with min-max normalization to limit values in each dimension
between 0 and 1(Hande et al., 2021).

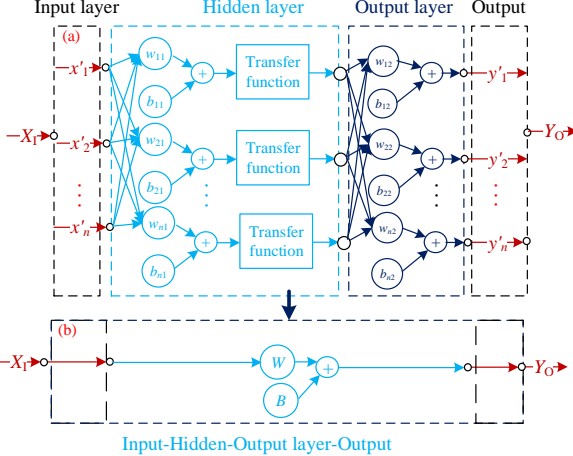

**Figure 4. Topological structure of BP neural network model. The up panel (a) is the feed-forward topological structure. The $X_I$ and $Y_O$**
**are the input data and output data, respectively. The $X'_i$ and $Y'_i$ separately indicate the normalized items of $X$ and $Y$. The $w_{i1}$ and $b_{i1}$, $w_{j2}$**
**and $b_{j2}$ separately represent the weight value and intercept value of the hidden layer and output layer. The down panel (b) is equivalent**
**to panel (a) to simplify the formulas.**
After the normalization process, the BP network can be established. To optimize the best parameters of the network, the number
of hidden layer, the transfer functions of the layers, and the end conditions should be determined. If the parameters are inappropriate,
the BP-model will be over trained or insufficient. In this study, a shallow structure with a single hidden layer is chosen, as extensive
testing did not show any noticeable improvement in calibration performance with deeper structure consisting of multiple hidden
layers(Ali et al., 2021). This also reduced the complexity and the training time.
**3.1.3 Genetic algorithm-BP model**
In the traditional BP neural network, the initial weights and thresholds are randomly generated. The results often fall into a local
minimum rather than a global minimum, and would lead to the distortion of the prediction result. In addition, the convergence
speed of the BP neural network is usually slow. To solve these problems, the genetic algorithm (GA) (Liang et al., 2018) with BP
algorithm is also used to avoid the inherent defects of BP algorithm. The GA method is essentially a direct search method that does
not rely on specific problems and gradient information. It follows the survival and elimination rule of biological evolution,
generates the following hypotheses by mutating and reconstructing the best existed hypothesis and makes it possible to solve the
problem(Ning et al., 2019). Generally, the GA is used to find an optimal initial weight and a threshold value for the model, so that
the model could converge in the direction of minimum value(Wang et al., 2019). The GA-BP hybrid algorithm is used to reduce
the time for the BP neural network to adjust the weight and threshold itself and achieve the goal of improving work efficiency.
**3.1.4 K nearest neighbor model**
The $k$ nearest neighbor (KNN) is also one of the simplest method for classification as well as regression problem(Kumar, 2015;
Zhao et al., 2021). The KNN is a supervised method that uses estimation based on values of neighbors, which can automatically
adapt to the supervised learning problems with arbitrary Bayes decision boundaries(Zhao et al., 2021). From the supervisor dataset,
the KNN solution utilizes the values of given dependent variable $y_i$ to approximate the dependent variable $y^*$, which is close with
respect to distance between their corresponding model parameters. For regression problem, the mean of the observed labels of $k$
nearest neighbors of independent variable $X$ is assigned to be the predicted label. In this study, the $k$ is set to 10 with the performance
having no obvious difference from other numbers.
**3.1.5 Random forest model**
The random forest (RF) model is used for solving regression or classification problems(Breiman, 2001; Liu et al., 2012). It works
by constructing an ensemble of decision trees using a training data set; the mean value from that ensemble of decision trees is then
used to predict the value for new input data(Zimmerman et al., 2018). With the purpose to establish a RF model, the maximum
number of decision trees of the forest should be specified. Each tree is constructed using a bootstrapped random sample from the
training data set. By considering a random subset of the possible explanatory variables with the strongest predictor of the response,
the origin node of the decision tree can be split into sub-nodes.  The node splitting process is repeated until a terminal node is
reached. The terminal node can be specified using the maximum number of sub-nodes or the minimum number of data points in
the node. To illustrate the method, consider building a random forest model for one LCS using a single decision tree and a subset
of 20490 data points to build a calibration model, shown in Figure 5. The RF model can predict data with variable parameters
within the training range. Therefore, a larger and more variable training data set should create a better final model. To avoid missing
any spikes during the training window, a 5-fold cross-validation approach(Zimmerman et al., 2018) is also used to maximize
utilization of the training data set. This approach helps to minimize bias in training data selection when predicting new data and
ensures that every point in the training window is used to build the model.

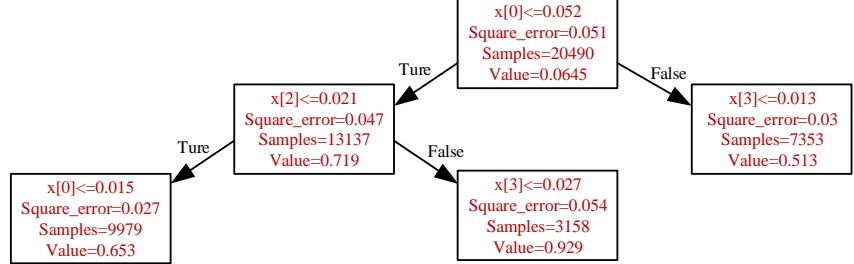

**Figure 5. Simplified illustration of the RF with a single decision tree and a subset. The $x[0]$, $x[2]$, $x[3]$ represent the CO, SO$_2$ and O$_3$**
**pollutants. At the first split, points with normalized CO sensor signal ≤0.052 are sent to a terminal node; the remaining points go to the**
**other splitting node. The Samples is the number of data points in each terminal node. The Value is the average in each terminal node.**
### 3.2 Metrics for performance evaluation
To quantitatively compare the performances of the five calibration models applied to the LCS, and balance the disadvantages of
the different metrics, the determination coefficient ($R^2$), root mean square error (RMSE) (Janabi et al., 2021), mean square error
(MSE) and mean absolute error (MAE) are utilized. The $R^2$ reflects the fit degree between the model output data and the reference
monitor measurement. The measurement results should meet the requirements of environmental standards of China(Wan et al.,
2016). The RMSE measures how much error there is between the predicted values and the reference measurements, and is sensitive
to extreme values(Chai et al., 2014). The MAE is a good choice to evaluate the error when the distribution is not Gaussian (Reza
et al., 2023). The formulas for the evaluation metrics are presented as equations (5) - (8), respectively.
$$R^2 = 1 - [\sum_{i=1}^{n} (y_i - \hat{y}_i)^2] / [\sum_{i=1}^{n} (y_i - \bar{y}_i)^2], \tag{5}$$
$$\text{RMSE} = \sqrt{\frac{1}{n}\sum_{i=1}^{n} (y_i - \hat{y}_i)^2}, \tag{6}$$
$$\text{MSE} = \frac{1}{n}\sum_{i=1}^{n} (y_i - \hat{y}_i)^2, \tag{7}$$
$$\text{MAE} = \frac{1}{n}\sum_{i=1}^{n} |y_i - \hat{y}_i|, \tag{8}$$
Where $\hat{y}_i$, $y_i$ and $\bar{y}$ represent the $i$th model output data form the algorithm-based LCS system, the reference data from the reference
instrumentations, and the mean value of the reference instrumentations, respectively. The $n$ is the number of the measurement data
in the dataset.
### 4 Results and discussion
Following the model building, the goodness of regression and root mean square error between the model output concentrations
and the reference monitor concentrations are evaluated for all calibration model approaches. The plots for the PM$_{2.5}$, PM$_{10}$, O$_3$, CO,
NO$_2$ and SO$_2$ illustrating the time series and goodness of fit of the models are provided in the Figure 6 - Figure 15. The $R^2$ and
RMSE values are listed in Table 2 - Table 9.
### 4.1 Parameters of the model
For the BP and GA-BP models, the parameters are the functions for the hidden layer and output layer, the type of the hidden layer,
the number of iteration times, and the number of the nerve units(Xu et al., 2021). The functions for the hidden layer and output
layer in this study respectively are the default tansig and the purelin functions. With the more complex type of the hidden layer
and less obvious improvement, the hidden layer is single type to achieve the goal of work efficiency.
To determine the best number of iteration times and nerve units, the measurement from the LCS and reference monitor between 1
March 2021 and 30 June 2021 is used. The number of iteration time is optimized using the mean squared error (MSE) between the
model value from the model and the reference monitor output value. The tendency of the MSE is shown in Figure 6. The training
is performed for 500 iterations. It is observed that the MSE decreases with the number of iteration time increasing, the rate of
decrease and the variation of the MSE is negligible beyond 100 iterations. More iterations incur higher computational cost for the
training and small performance improvement. There is also the risk of overtraining resulting in poor generalization capability.
Using this method, the same number of iteration times can be obtained with the different gas pollutants within 1 July 2021 and 31
October 2021, 1 November 2021 and 28 February 2022.

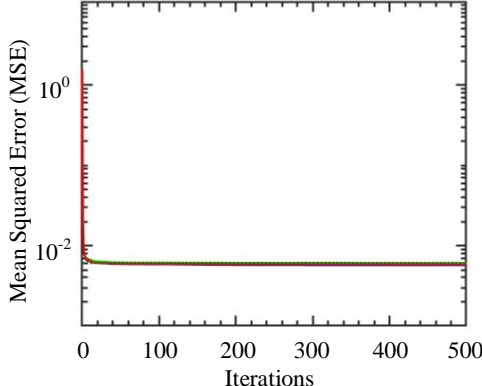

**Figure 6. The MSE with the number of iterations.**

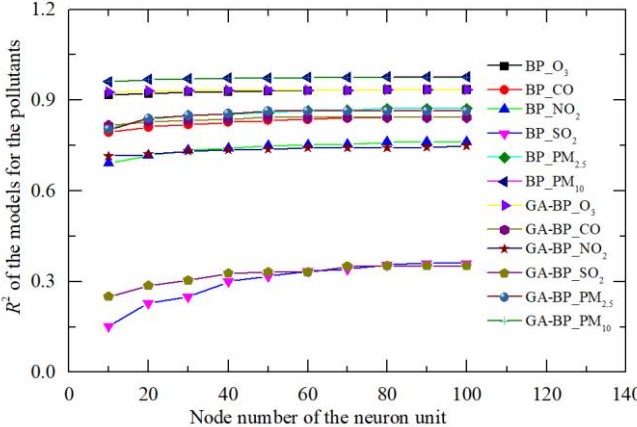

**Figure 7. The $R^2$ with different node number of the neuron for the pollutants.**
The node number of the nerve units is determined by the contrast results of determination coefficient $R^2$ for different gas and PM
pollutants within 1 March 2021 and 30 June 2021. The results are shown in Figure 7. The $R^2$ is improved as the number of nerve
units increasing. The rate of increase and the variation of $R^2$ is negligible beyond 70 units. More units incur higher computational
cost and time for the training and small performance improvement. Using this method, the same number of the nerve units can be
obtained with the different gas pollutants within 1 July 2021 and 31 October 2021, 1 November 2021 and 28 February 2022.
For the RF model, the number of trees is determined by using grid search method, which will search the optimal hyper-parameter
by traversing a given hyper-parameter combination (Zhu et al., 2022). A total of 11 kinds of tree numbers are set between 2 and
22. By using grid search to traverse these 11 kinds of tree numbers to obtain different $R^2$. For instance, the $R^2$ for different gas
pollutants within 1 March 2021 and 30 June 2021 are shown in Figure 8. The $R^2$ is improved as the number of trees increasing.
The rate of increase and the variation of $R^2$ is negligible beyond 20. The terminal node is specified using a maximum number of
sub-node points per node. The $R^2$ is also improved as the number of sub-node increasing under the same tree number. The rate of
increase and the variation of $R^2$ is negligible beyond 100. More number of the tree or the sub-node incur higher computational cost
and time for the training and small performance improvement. Using this method, the same number of trees can be obtained with
the different gas pollutants within 1 July 2021 and 31 October 2021, 1 November 2021 and 28 February 2022.

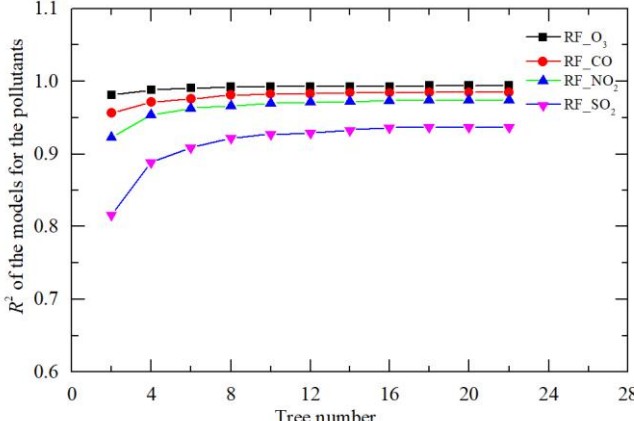

**Figure 8. The $R^2$ with different tree number of the RF model for the pollutants.**
**4.2  Measurement results of PM**
With the purpose of avoiding over-fit in the five models, the random partition parameters of train ratio and test ratio are 80% and
20%, respectively. To ensure the robustness of the model evaluation, the 5-fold cross validation is also conducted. The dataset is
divided into 5 mutually exclusive subsets with same size, where the 4 subsets are randomly selected as the training set each time,
and the remaining 1 subset is used as the test set. After completing each round of validation, 4 copies are selected again to train
the model and the remaining 1 copy is used for validation. After several rounds (less than 5), the loss function is selected to evaluate
the optimal model and parameters (Mahesh et al., 2023; Zimmerman et al., 2018).
With the results from 1 March 2021 to 28 February 2022 and according the trend of the ambient temperature, the whole data is
divided into three segments. The three segments (I, II, and III) separately are within 1 March 2021 and 30 June 2021 with the size
of 32481, 1 July 2021 and 31 October 2021 with the size of 31287, 1 November 2021 and 28 February 2022 with the size of 32053,
respectively.
During the measurement period, the ranges of the ambient temperature and relative humidity separately were minus 5℃ to plus
50℃ and 10% to 98%., shown in Figure 9. The ambient temperature raised, dropped and fluctuated separately within 1 March
2021 and 30 June 2021, 1 July 2021 and 31 October 2021, 1 November 2021 and 28 February 2022. The time series data and
regressions of five modes for PM from reference monitor and LCS calibration output are shown in Figure 10 and Figure 11.

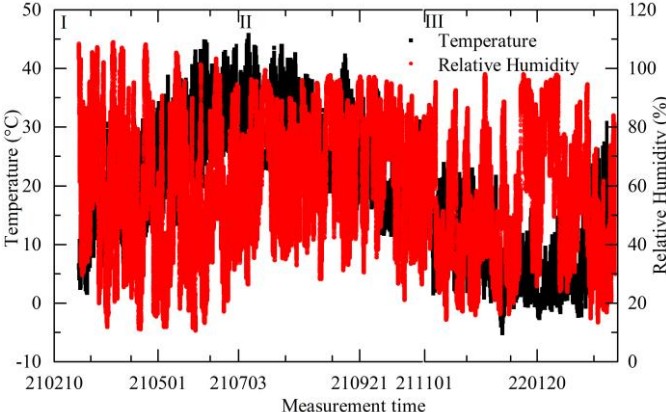

Figure 9 Temperature/relative humidity ranges during the measurement period.

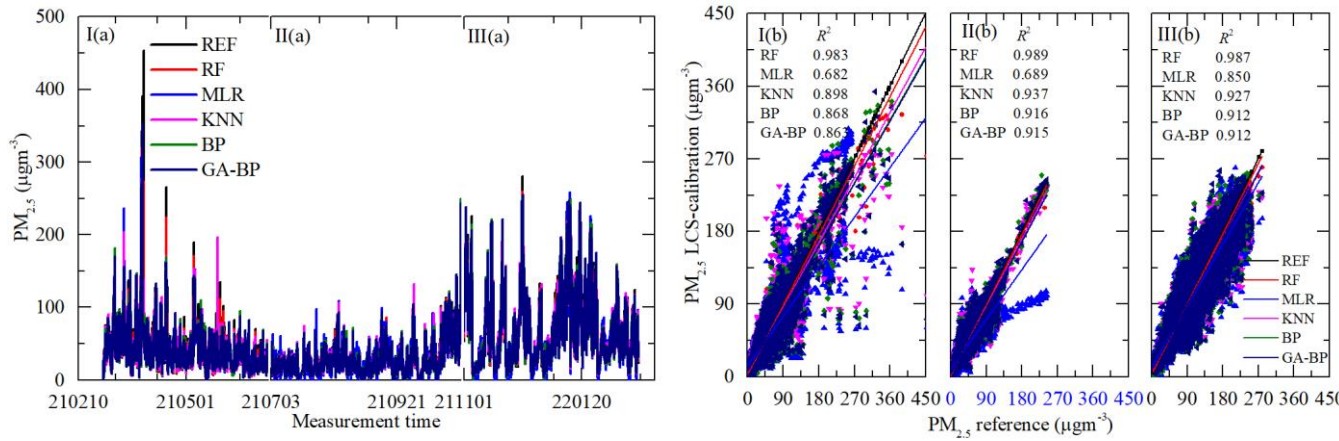

Figure 10. Time series and regressions comparing the reference monitor PM$_{2.5}$ data (black) to five calibration model PM$_{2.5}$ results. Where red, blue, magenta, olive and navy represent RF, MLR, KNN, BP, GA-BP, respectively. The left panel (a) shows the whole time series data of the measurement period. The right panel (b) shows the regressions of the five calibration models.

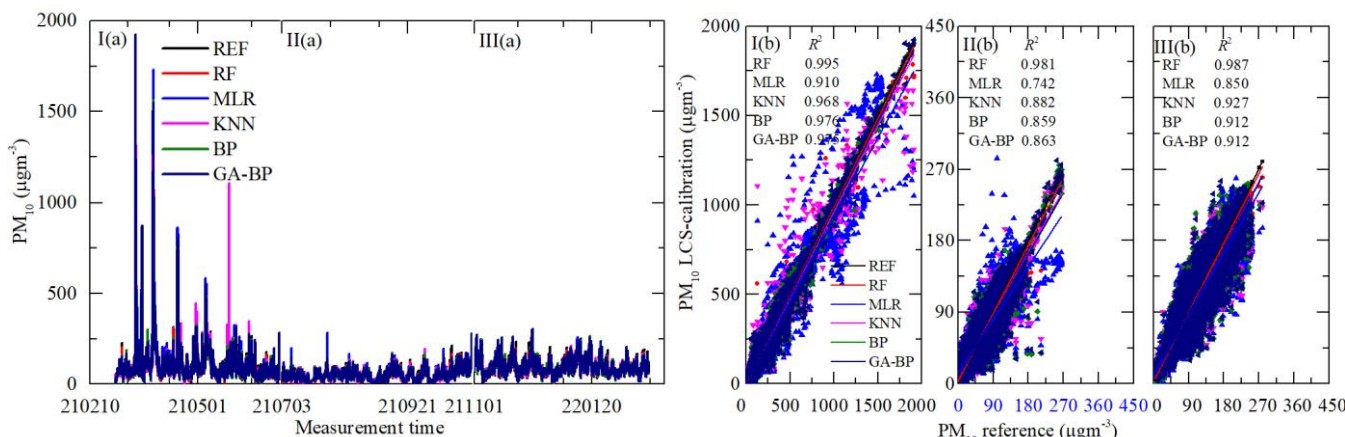

Figure 11. Time series and regressions comparing the reference monitor PM$_{10}$ data (black) to five calibration model PM$_{10}$ results. Where red, blue, magenta, olive and navy represent RF, MLR, KNN, BP, GA-BP, respectively. The left panel (a) shows the whole time series data of the measurement period. The right panel (b) shows the regressions of the five calibration models.

As shown in Figure 10(a) and Figure 11(a), the general tendency of the data fluctuation between the reference monitor and the RF, MLR, KNN, BP, GA-BP based algorithm of the LCS are consistent with each other. The best performance is RF model, the next are KNN, BP and GA-BP, the worst is MLR.

The $R^2$ between the reference data and the five model data are also shown in Figure 10(b) and Figure 11(b), and listed in Table 2. The $R^2$ of RF for the PM is better than 0.98. The $R^2$ of MLR for the PM is less than 0.91, and even less than 0.7. The $R^2$ of the other three model are within 0.86 and 0.98.

**Table 2. Performance of different calibration models for the PM$_{2.5}$ and PM$_{10}$ against reference monitor. The determination coefficient $R^2$ (higher is better, maximum of 1) of different calibration models (RF, MLR, KNN, BP, GA-BP) versus reference monitor.**

| $R^2$ / Model | PM$_{2.5}$ | | | PM$_{10}$ | | |
|---|---|---|---|---|---|---|
| | I | II | III | I | II | III |
| RF | **0.983** | **0.989** | **0.987** | **0.995** | **0.981** | **0.987** |
| MLR | 0.682 | 0.689 | 0.850 | 0.910 | 0.742 | 0.850 |
| KNN | 0.898 | 0.937 | 0.927 | 0.968 | 0.882 | 0.927 |
| BP | 0.868 | 0.916 | 0.912 | 0.976 | 0.859 | 0.912 |
| GA-BP | 0.863 | 0.915 | 0.912 | 0.975 | 0.863 | 0.912 |

The performance of different calibration models for the PM against reference monitor is also evaluated using RMSE, MSE and MAE. The results are listed in Table 3, Table 4, and Table 5, respectively. Using the data listed in Table 3, the RMSE values from the first (I) and third (III) periods are large than the one from the second (II) stage, the main reason maybe the large fluctuation range of the PM for the climatic factors in winter and spring resulting in the poor model fit. The RMSE values of PM$_{2.5}$ between the reference data and the RF, MLR, KNN, BP, GA-BP-based algorithms data are within 2.36 - 5.49, 12.63 - 18.68, 5.67 - 13.05,6.56 - 14.35, 6.61 - 14.35, respectively. The RMSE values of PM$_{10}$ between the reference data and the RF, MLR, KNN, BP, GA-BP-based algorithms data are calculated as 4.55 -10.37, 16.43 - 45.05, 11.14 - 27.08, 12.15 - 23.10, and 11.99 - 23.65, respectively.

**Table 3. Performance of different calibration models for the PM$_{2.5}$ and PM$_{10}$ against reference monitor. The RMSE values (lower is better) of different calibration models (RF, MLR, KNN, BP, GA-BP) versus reference monitor.**

| RMSE / Model | PM$_{2.5}$ | | | PM$_{10}$ | | |
|---|---|---|---|---|---|---|
| | I | II | III | I | II | III |
| RF | 4.03 | 2.36 | 5.49 | 10.37 | 4.55 | 7.19 |
| MLR | 17.18 | 12.63 | 18.68 | 45.05 | 16.43 | 25.22 |
| KNN | 9.73 | 5.67 | 13.05 | 27.08 | 11.14 | 17.29 |
| BP | 11.09 | 6.56 | 14.35 | 23.10 | 12.15 | 18.88 |
| GA-BP | 11.27 | 6.61 | 14.35 | 23.65 | 11.99 | 18.87 |

Using the data listed in Table 4 and Table 5, the MSE and MAE values have the same characteristics with RMSE. The MSE values of PM$_{2.5}$ between the reference data and the RF, MLR, KNN, BP, GA-BP-based algorithms data are within 5.58 -30.12, 159.52 - 348.84, 32.14 - 170.37, 43.07 - 205.90, and 43.63 - 205.83, respectively. The MSE values of PM$_{10}$ between the reference data and the RF, MLR, KNN, BP, GA-BP-based algorithms data are within 20.70 -107.47, 270.07 - 2029.23, 124.06 - 733.49, 147.60 - 533.44, and 143.66 - 559.20, respectively.

**Table 4. Performance of different calibration models for the PM$_{2.5}$ and PM$_{10}$ against reference monitor. The MSE values (lower is better) of different calibration models (RF, MLR, KNN, BP, GA-BP) versus reference monitor.**

| MSE / Model | PM$_{2.5}$ | | | PM$_{10}$ | | |
|---|---|---|---|---|---|---|
| | I | II | III | I | II | III |
| RF | 16.24 | 5.58 | 30.12 | 107.47 | 20.70 | 51.68 |
| MLR | 295.25 | 159.52 | 348.84 | 2029.23 | 270.07 | 636.04 |
| KNN | 94.59 | 32.14 | 170.37 | 733.49 | 124.06 | 298.86 |
| BP | 123.00 | 43.07 | 205.90 | 533.44 | 147.60 | 356.42 |
| GA-BP | 126.92 | 43.63 | 205.83 | 559.20 | 143.66 | 355.97 |

The MAE values of PM$_{2.5}$ between the reference data and the RF, MLR, KNN, BP, GA-BP-based algorithms data are within 1.44 - 3.45, 8.37 - 12.80, 3.56 - 8.31, 4.46 - 9.55, and 4.48 - 9.54, respectively. The MAE values of PM$_{10}$ between the reference data and

the RF, MLR, KNN, BP, GA-BP-based algorithms data are within 3.21 - 5.28, 12.21 - 23.20, 8.00 - 13.35, 8.99 - 15.26, and 8.89 - 15.43, respectively.

**Table 5. Performance of different calibration models for the PM$_{2.5}$ and PM$_{10}$ against reference monitor. The MAE values (lower is better) of different calibration models (RF, MLR, KNN, BP, GA-BP) versus reference monitor.**

| MAE / Model | PM$_{2.5}$ | | | PM$_{10}$ | | |
|---|---|---|---|---|---|---|
| | I | II | III | I | II | III |
| RF | 2.19 | 1.44 | 3.45 | 5.28 | 3.21 | 5.13 |
| MLR | 10.92 | 8.37 | 12.80 | 23.20 | 12.21 | 19.10 |
| KNN | 5.55 | 3.56 | 8.31 | 13.35 | 8.00 | 12.31 |
| BP | 7.34 | 4.46 | 9.55 | 15.26 | 8.99 | 14.06 |
| GA-BP | 7.42 | 4.48 | 9.54 | 15.43 | 8.89 | 14.07 |

## 4.3  Measurement results of gas pollution

With the results from 1 March 2021 to 28 February 2022 and according the trend of the ambient temperature, shown in Figure 9, the whole data is also divided into three same segments as section 4.2. With the same purpose of avoiding over-fit in the five models and ensure the robustness of the model evaluation, the random partition parameters of train ratio and test ratio are also 80% and 20%, and the 5-fold cross validation is also conducted. The time series data and regressions of five modes for gas pollution from reference monitor and LCS calibration output are shown in Figure 12 - Figure 15.

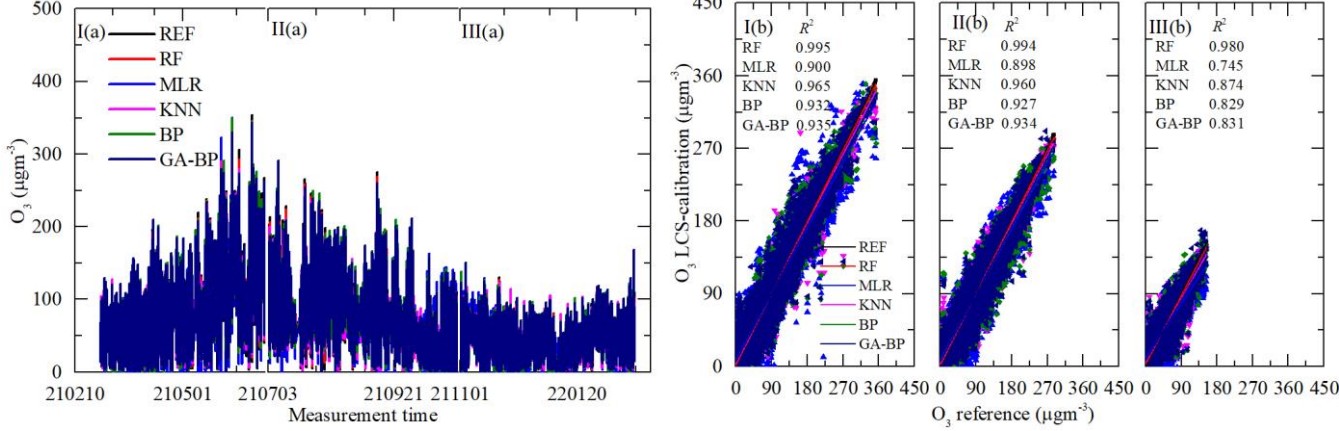

**Figure 12. Time series and regressions comparing the reference monitor O$_3$ data (black) to five calibration model O$_3$ results. Where red, blue, magenta, olive and navy represent RF, MLR, KNN, BP, GA-BP, respectively. The left panel (a) shows the whole time series data of the measurement period. The right panel (b) shows the regressions of the five calibration models.**

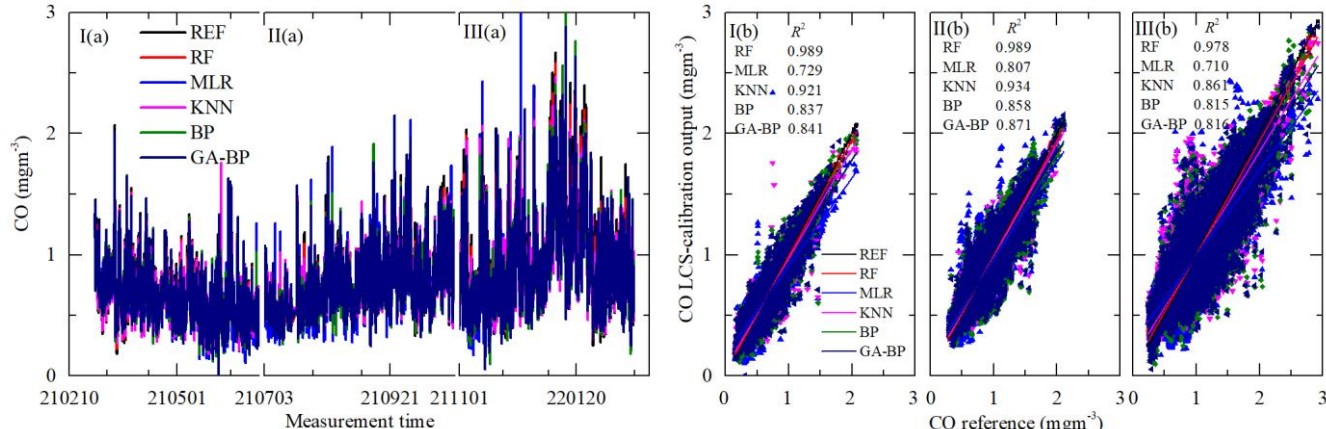

**Figure 13. Time series and regressions comparing the reference monitor CO data (black) to five calibration model CO results. Where red, blue, magenta, olive and navy represent RF, MLR, KNN, BP, GA-BP, respectively. The left panel (a) shows the whole time series data of the measurement period. The right panel (b) shows the regressions of the five calibration models.**

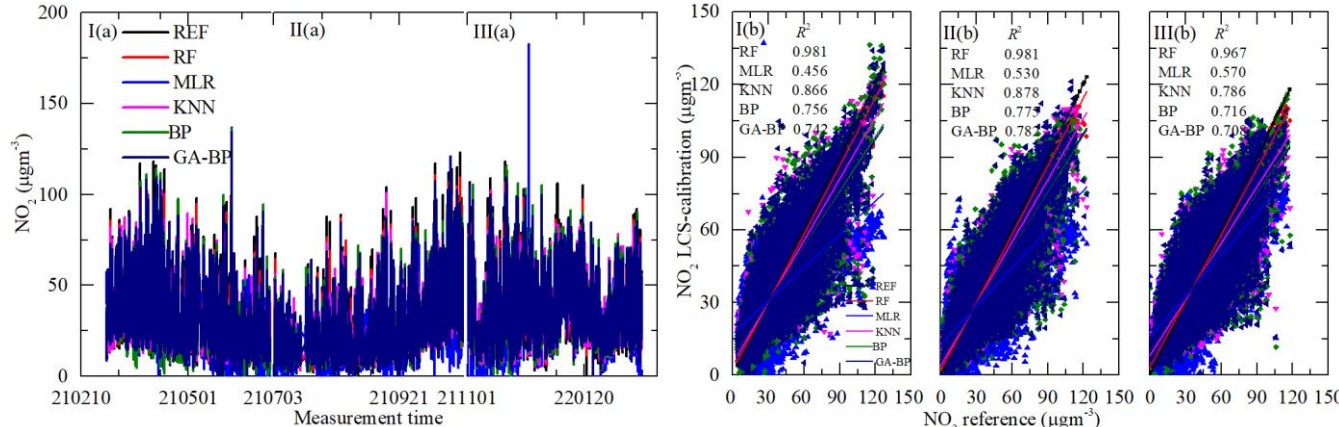

Figure 14. Time series and regressions comparing the reference monitor NO₂ data (black) to five calibration model NO₂ results. Where
red, blue, magenta, olive and navy represent RF, MLR, KNN, BP, GA-BP, respectively. The left panel (a) shows the whole time series
data of the measurement period. The right panel (b) shows the regressions of the five calibration models.

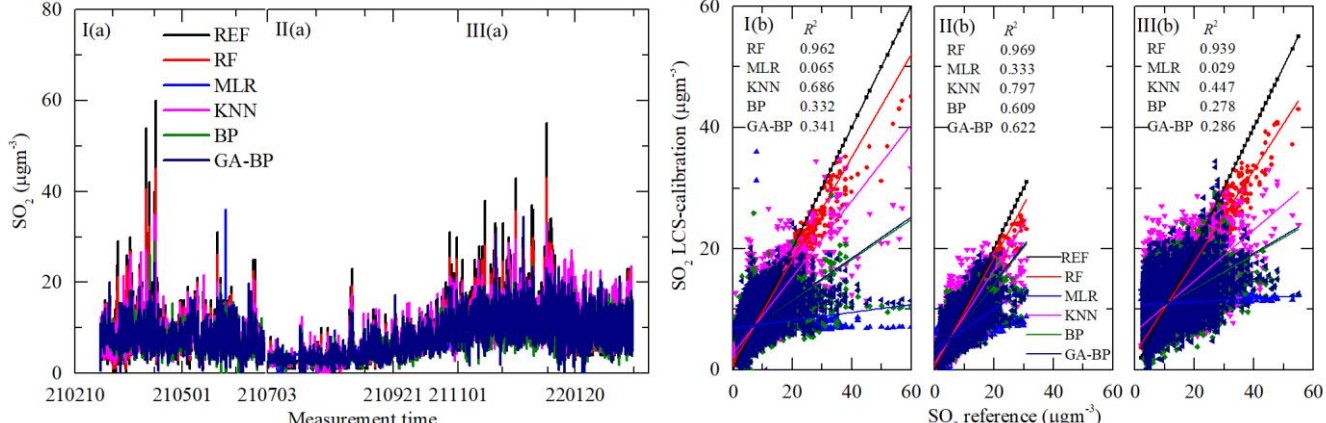

Figure 15. Time series and regressions comparing the reference monitor SO₂ data (black) to five calibration model SO₂ results. Where
red, blue, magenta, olive and navy represent RF, MLR, KNN, BP, GA-BP, respectively. The left panel (a) shows the whole time series
data of the measurement period. The right panel (b) shows the regressions of the five calibration models.
As shown in Figure 12(a) - Figure 15(a), the general tendency of the data fluctuation between the reference monitor and the RF,
MLR, KNN, BP, GA-BP based algorithm of the LCS are consistent with each other. The best performance is RF model, the next
are KNN, BP and GA-BP, the worst is MLR. The $R^2$ between the reference data and the five model data are also shown in Figure
12(b) - Figure 15(b), and listed in Table 6.
**Table 6. Performance of different calibration models for the gaseous pollutant (SO₂, CO, NO₂, and O3) against reference monitor. The**
**determination coefficient $R^2$ (higher is better, maximum of 1) of different calibration models (RF, MLR, KNN, BP, GA-BP) versus**
**reference monitor.**

| $R^2$ / Model | O₃ | | | CO | | | NO₂ | | | SO₂ | | |
|---|---|---|---|---|---|---|---|---|---|---|---|---|
| | I | II | III | I | II | III | I | II | III | I | II | III |
| RF | **0.995** | **0.994** | **0.980** | **0.989** | **0.989** | **0.978** | **0.981** | **0.981** | **0.967** | **0.962** | **0.969** | **0.939** |
| MLR | 0.900 | 0.898 | 0.745 | 0.729 | 0.807 | 0.710 | 0.456 | 0.530 | 0.570 | 0.065 | 0.333 | 0.029 |
| KNN | 0.965 | 0.960 | 0.874 | 0.921 | 0.934 | 0.861 | 0.866 | 0.878 | 0.786 | 0.686 | 0.797 | 0.447 |
| BP | 0.932 | 0.927 | 0.829 | 0.837 | 0.858 | 0.815 | 0.756 | 0.775 | 0.716 | 0.332 | 0.609 | 0.278 |
| GA-BP | 0.935 | 0.934 | 0.831 | 0.841 | 0.871 | 0.816 | 0.742 | 0.782 | 0.708 | 0.341 | 0.622 | 0.286 |

For the O₃ model, the $R^2$ of RF is better than 0.98. The $R^2$ of MLR is less than 0.90, and even less than 0.8. The $R^2$ of the other
three models are within 0.82 and 0.97. For the CO model, the $R^2$ of RF is better than 0.97. The $R^2$ of MLR is less than 0.81, and
even less than 0.7. The $R^2$ of other three models are within 0.81 and 0.94. For the NO₂ model, the $R^2$ of RF is better than 0.96. The
$R^2$ of MLR is less than 0.60, and even less than 0.5. The $R^2$ of other three models are within 0.70 and 0.90. For the SO₂ model, the

$R^2$ of RF is better than 0.93. The $R^2$ of MLR is less than 0.40, and even less than 0.1. The $R^2$ of other three models are within 0.27 and 0.80.

The performances of different calibration models for the gas pollution against reference monitor are also evaluated using RMSE, MSE and MAE. The results are listed in Table 7, Table 8, and Table 9, respectively.

**Table 7. Performance of different calibration models for the gaseous pollutant ($SO_2$, CO, $NO_2$ and $O_3$) against reference monitor. The RMSE values (lower is better) of different calibration models (RF, MLR, KNN, BP, GA-BP) versus reference monitor.**

| RMSE Model | $O_3$ | | | CO | | | $NO_2$ | | | $SO_2$ | | |
|---|---|---|---|---|---|---|---|---|---|---|---|---|
| | I | II | III | I | II | III | I | II | III | I | II | III |
| RF | 4.05 | 4.06 | 4.08 | 0.02 | 0.03 | 0.06 | 2.88 | 2.88 | 3.99 | 0.83 | 0.64 | 1.68 |
| MLR | 17.79 | 16.42 | 14.00 | 0.12 | 0.12 | 0.23 | 14.54 | 13.54 | 13.61 | 3.53 | 2.69 | 5.37 |
| KNN | 10.57 | 10.28 | 9.84 | 0.06 | 0.07 | 0.16 | 7.25 | 6.93 | 9.61 | 2.06 | 1.49 | 4.05 |
| BP | 14.67 | 13.91 | 11.46 | 0.09 | 0.10 | 0.18 | 9.75 | 9.37 | 11.07 | 2.98 | 2.06 | 4.63 |
| GA-BP | 14.40 | 13.19 | 11.41 | 0.09 | 0.10 | 0.18 | 10.02 | 9.21 | 11.21 | 2.97 | 2.03 | 4.60 |

Using the data listed in Table 7, the RMSE values of $O_3$, CO and $NO_2$ from the first (I) and third (III) periods have little difference with the one from the second (II) period, indicating the $O_3$, CO and $NO_2$ electrochemical sensor suitable for the ambient $O_3$, CO and $NO_2$ measurement. The RMSE values of $O_3$ between the reference data and the RF, MLR, KNN, BP, GA-BP-based algorithms data are within 4.05 - 4.08, 14.00 - 17.79, 9.84 - 10.57, 11.46 - 14.67, and 11.41 - 14.40, respectively. The RMSE values of CO between the reference data and the RF, MLR, KNN, BP, GA-BP-based algorithms data are within 0.02 - 0.06, 0.12 - 0.23, 0.06 - 0.16, 0.09 - 0.18, and 0.09 - 0.18, respectively. The RMSE values of $NO_2$ between the reference data and the RF, MLR, KNN, BP, GA-BP-based algorithms data are within 2.88 - 3.99, 13.54 - 14.54, 6.93 - 9.61, 9.37 - 11.07, and 9.21 - 11.21, respectively.

Using the RF model, the RMSE values of $SO_2$ are better than the values of other methods, but still have difference during the three periods. However, using other models, the RMSE values of $SO_2$ from the first (I) and third (III) periods are larger than the one from the second (II) period, the main reason maybe the large ambient fluctuation for the climatic factors in winter and spring, resulting in the poor model fit. The RMSE values of $SO_2$ between the reference data and the RF, MLR, KNN, BP, GA-BP-based algorithms data are within 0.64 - 1.68, 2.69 - 5.37, 1.49 - 4.05, 2.06 - 4.63, and 2.03 - 4.60, respectively.

Using the data listed in Table 8 and Table 9, the MSE and MAE values have the same characteristics with RMSE. The MSE values of $O_3$ between the reference data and the five algorithm-based data are within 16.43 -16.68, 196.02 - 316.40, 96.77 - 111.70, 131.33 - 215.14, and 130.20 - 207.37, respectively. The MSE values of CO between the reference data and the RF, MLR, KNN, BP, GA-BP-based algorithms data are within $5.97 \times 10^{-4}$ - $4.21 \times 10^{-3}$, $1.38 \times 10^{-2}$ - $5.13 \times 10^{-2}$, $4.02 \times 10^{-3}$ - $2.46 \times 10^{-2}$, $8.29 \times 10^{-3}$ - $3.27 \times 10^{-2}$, and $8.09 \times 10^{-3}$ - $3.26 \times 10^{-2}$, respectively. The MSE values of $NO_2$ between the reference data and the RF, MLR, KNN, BP, GA-BP-based algorithms data are within 8.27 - 15.91, 183.36 - 211.48, 48.01 - 92.39, 87.76 - 122.55, and 84.90 - 125.73, respectively. The MSE values of $SO_2$ between the reference data and the RF, MLR, KNN, BP, GA-BP-based algorithms data are within 0.41 - 2.84, 7.24 - 28.80, 2.22 - 16.44, 4.25 - 21.39, and 4.11 - 21.16, respectively.

**Table 8. Performance of different calibration models for the gaseous pollutant ($SO_2$, CO, $NO_2$ and $O_3$) against reference monitor. The MSE values (lower is better) of different calibration models (RF, MLR, KNN, BP, GA-BP) versus reference monitor.**

| MSE Model | $O_3$ | | | CO | | | $NO_2$ | | | $SO_2$ | | |
|---|---|---|---|---|---|---|---|---|---|---|---|---|
| | I | II | III | I | II | III | I | II | III | I | II | III |
| RF | 16.43 | 16.45 | 16.68 | $5.97 \times 10^{-4}$ | $9.22 \times 10^{-4}$ | $4.21 \times 10^{-3}$ | 8.27 | 8.28 | 15.91 | 0.68 | 0.41 | 2.84 |
| MLR | 316.40 | 269.65 | 196.02 | $1.38 \times 10^{-2}$ | $1.39 \times 10^{-2}$ | $5.13 \times 10^{-2}$ | 211.48 | 183.36 | 185.27 | 12.49 | 7.24 | 28.80 |
| KNN | 111.70 | 105.59 | 96.77 | $4.02 \times 10^{-3}$ | $4.81 \times 10^{-3}$ | $2.46 \times 10^{-2}$ | 52.49 | 48.01 | 92.39 | 4.26 | 2.22 | 16.44 |
| BP | 215.14 | 193.62 | 131.33 | $8.29 \times 10^{-3}$ | $1.02 \times 10^{-2}$ | $3.27 \times 10^{-2}$ | 95.07 | 87.76 | 122.55 | 8.91 | 4.25 | 21.39 |
| GA-BP | 207.37 | 174.05 | 130.20 | $8.09 \times 10^{-3}$ | $9.33 \times 10^{-3}$ | $3.26 \times 10^{-2}$ | 100.50 | 84.90 | 125.73 | 8.80 | 4.11 | 21.16 |

The MAE values of $O_3$ between the reference data and the RF, MLR, KNN, BP, GA-BP-based algorithms data are within 2.76 - 2.88, 10.79 - 13.46, 7.06 - 7.33, 8.70 - 11.14, and 8.67 - 10.90, respectively. The MAE values of CO between the reference data

and the RF, MLR, KNN, BP, GA-BP-based algorithms data are within 0.02 - 0.05, 0.09 - 0.19, 0.04 - 0.11, 0.07 - 0.14, and
0.07 - 0.14, respectively. The MAE values of $NO_2$ between the reference data and the RF, MLR, KNN, BP, GA-BP-based
algorithms data are within 1.84 - 2.80, 10.41 - 11.08, 4.45 - 6.85, 6.59 - 8.27, and 6.48 - 8.41, respectively. The MAE values of
$SO_2$ between the reference data and the RF, MLR, KNN, BP, GA-BP-based algorithms data are within 0.39 - 1.16, 1.96 - 4.24,
0.91 - 2.84, 1.41 - 3.43, and 1.36 - 3.40, respectively.
**Table 9. Performance of different calibration models for the gaseous pollutant ($SO_2$, CO, $NO_2$ and $O_3$) against reference monitor. The**
**MAE values (lower is better) of different calibration models (RF, MLR, KNN, BP, GA-BP) versus reference monitor.**

| MAE Model | $O_3$ | | | CO | | | $NO_2$ | | | $SO_2$ | | |
|---|---|---|---|---|---|---|---|---|---|---|---|---|
| | I | II | III | I | II | III | I | II | III | I | II | III |
| RF | 2.76 | 2.83 | 2.88 | 0.02 | 0.02 | 0.05 | 1.86 | 1.84 | 2.80 | 0.49 | 0.39 | 1.16 |
| MLR | 13.46 | 12.77 | 10.79 | 0.09 | 0.09 | 0.19 | 11.08 | 10.41 | 10.74 | 2.54 | 1.96 | 4.24 |
| KNN | 7.33 | 7.22 | 7.06 | 0.04 | 0.04 | 0.11 | 4.74 | 4.45 | 6.85 | 1.25 | 0.91 | 2.84 |
| BP | 11.14 | 10.60 | 8.70 | 0.07 | 0.08 | 0.14 | 7.07 | 6.59 | 8.27 | 2.08 | 1.41 | 3.43 |
| GA-BP | 10.90 | 10.02 | 8.67 | 0.07 | 0.07 | 0.14 | 7.31 | 6.48 | 8.41 | 2.05 | 1.36 | 3.40 |

As shown in Figure 12 - Figure 15 and listed in Table 6 - Table 9, the results of each model have little difference among the three
periods for the $O_3$, CO and $NO_2$ measurement, and the RF model outperforms other models.
For the data of $SO_2$, the results of RF are better than the ones of other methods, and have little difference among the three periods.
However, the performances of other methods (MLR, KNN, BP, GA-BP) are poorer than the one during the first and third periods.
There may be some reasons for this phenomenon. The first one is the cross interference effect from $NO_2$ and $O_3$, which have the
wide range of fluctuations (from about 20 $\mu gm^{-3}$ to 125 $\mu gm^{-3}$) and increasing tendency in period I, respectively. The $NO_2$ and $SO_2$
can react chemically under certain conditions to produce sulfuric acid ($H_2SO_4$) and nitric acid ($HNO_3$), which will affect the reading
of $SO_2$ sensor. The $O_3$, highly oxidizing gas, may react with $SO_2$ to form $H_2SO_4$ or sulfite ($H_2SO_3$), resulting in inaccurate sensor
readings. The second one is the ambient temperature has a wide range of fluctuations (from about minus 5 ℃ to plus 45 ℃) during
the first and third periods, which will affect the stability of electrode material and the readings of the sensor. The last one is the
concentration of ambient $SO_2$ is high (more than 30 $\mu gm^{-3}$) in period I and period III, beyond the actual measurement range of the
$SO_2$ sensor, which will be researched in future.
**5  Conclusions and Discussion**
A low-cost air quality monitoring system (LCS) based on RF, MLR, KNN, BP, GA-BP algorithms are proposed. The system can
measure gas-phase pollutants ($SO_2$, $NO_2$, CO and $O_3$) and particle pollutants ($PM_{2.5}$ and $PM_{10}$), simultaneously. With the purpose
to estimate the performance of the five algorithms, the LCS was mounted at the same location (Zhengzhou City, China) and
consistent height with the reference monitoring system. The measurement was made continuously from 1 March 2021 to 28
February 2022, with the ranges of the ambient temperature and relative humidity separately minus 5℃ to plus 50℃ and 10% to
98%, respectively. The values of the LCS and reference instruments were separately logged to the server for further comparative
analysis.
With the pretreated and individual particle counters, $T$ and $RH$ as input, and the concentrations of $PM_{2.5}$ and $PM_{10}$ measured by the
reference instrumentation separately as output, the multi-input one-output evaluation models based on RF, MLR, KNN, BP, GA-
BP algorithms can be obtained. With the four types of electro-chemical sensors raw data, $T$ and $RH$ as input, and the measurements
from the reference monitors as output, the multi-input multi-output evaluation models based on the five algorithms can be obtained.
The performance of the calibration models are quantitatively compared by utilizing $R^2$, RMSE, MSE and MAE.
The experimental results show that the $R^2$ of RF for the PM is better than 0.98; the $R^2$ of MLR for the PM is less than 0.91; the $R^2$
of the other three model are within 0.86 and 0.98. The $R^2$ of RF for the gas pollutants ($SO_2$, $NO_2$, CO and $O_3$) is better than 0.93;

the $R^2$ of KNN, BP and GA-BP for the gas pollutants ($SO_2$, $NO_2$, CO and $O_3$) is within 0.27 to 0.97; the $R^2$ of MLR for the $NO_2$, CO and $O_3$ is within 0.46 to 0.90, but for SO2 less than 0.40, and even less than 0.1.

The maximum RMSE values of $PM_{2.5}$, $PM_{10}$, $O_3$, CO, $NO_2$, and $SO_2$ between the reference data and the RF, MLR, KNN, BP, GA-BP-based algorithms data are 5.49, 18.68, 13.05, 14.35, and 14.35; 10.37, 45.05, 27.08, 23.10, and 23.65; 4.08, 17.79, 10.57, 14.67, and 14.40; 0.06, 0.23, 0.16, 0.18, and 0.18; 3.99, 14.54, 9.61, 11.07, and 11.21; 1.68, 5.37, 4.05, 4.63, and 4.60, respectively. The maximum MAE values of $PM_{2.5}$, $PM_{10}$, $O_3$, CO, $NO_2$, and $SO_2$ between the reference data and the RF, MLR, KNN, BP, GA-BP-based algorithms data are 3.45, 12.80, 8.31, 9.55, and 9.54; 5.28, 23.20, 13.35, 15.26, and 15.43; 2.88, 13.46, 7.33, 11.14, and 10.90; 0.05, 0.19, 0.11, 0.14, and 0.14; 2.80, 11.08, 6.85, 8.27, and 8.41; 1.16, 4.24, 2.84, 3.43, and 3.40, respectively.

It is should be noted that the results of RF are better than the ones of other methods, have very good agreement with the reference monitors and little difference among the three periods. However, the performances of other methods (MLR, KNN, BP, GA-BP) have poor agreement, especially during the first and third periods. There may be some reasons, such as the cross interference effect, the wide range of fluctuation of the climatic factors, and the limitation of the actual measurement range and precision.

Overall, we conclude that, with careful data management and calibration using the machine learning algorithms, especially the RF method, these measurements are consistent with the national environmental protection standard requirement of China, the LCS may significantly improve our ability to spatial heterogeneity in air pollutant concentrations. The air pollutant maps will assist researchers, policymakers, and communities in developing new policies or mitigation strategies to enhance human health. In the next research, we will focus on improving the matching of the measurement precise and range, the generalization of the algorithms in more applications, and the performance of the $SO_2$ sensor.

**Competing interests**

The contact author has declared that none of the authors has any competing interests.

**Acknowledgment**

Funding for this study was provided by the National Key Research and Development Program of China (Assistance Agreement No. 2021YFB3200403) and Zhengzhou Education Department (No. 23B413006). The authors also wish to thank Minghui, Li, Hongbiao Liu, and Jinlong Wang for helpful conversations.

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
