# Peer review of "Research of Low-cost Air Quality Monitoring Models with Different"

_Atmospheric Measurement Techniques, 2023_

## Author Comment (AC2)

**RC1:**

**This is a fine piece of measurement calibration paper. My understanding on measurement techniques are very thin, so I will only comment on data processing:**

**(1) From my point of view, the authors applied different algorithms to fit the low cost sensor data and compared the performance, but the authors do not discuss how the results can be used to predict the PM and other pollutants. Specifically, is it used to predict unobserved locations or forecast short-term future?**

Thanks for your question. You are right that we applied different algorithms to fit the low-cost sensor data and compared the performance. **The prediction** in this paper means that we use the fitted model from the empirical value to estimate the current raw data from the sensor with the same location, with the purpose to get more accurate result. The model used to predict unobserved locations or forecast short-term future will be discussed in the future research.

**(2) Also, let's say, RF seems to outperform other methods, is RF calibrated output considered to be the final product? or low cost sensor raw data still should be the reference?**

Thanks for your question. The RF outperforms other methods. The calibrated output result of RF in this paper is accurate enough, and can be the final product. The current raw data of the sensor is still used as the input of the model to obtain the current accurate calibrated data.

**(3)Do the authors consider the better performance from RF that can be overfitting, and other methods with a lower predictive performance can be more explainable?**

Thanks for your question. With the purpose of avoiding over-fit in the five models, the randomly divide parameters of train ratio and test ratio are 80% and 20%, respectively. To ensure the robustness of the model evaluation, the 5-fold cross validation is also conducted. The dataset is divided into 5 mutually exclusive subsets with same size, the 4 subset is randomly selected as the training set each time, and the remaining 1 subset is used as the test set. After completing each round of validation, select 4 copies again to train the model and use the remaining 1 copy for validation. After several rounds (less than 5), the loss function is selected to evaluate the optimal model and parameters (Mahesh et al., 2023; Zimmerman et al., 2018).

**(4) The formula in Equation (1) does not seem valid for MLR, because all those terms are correlated. Additional treatments or justifications are needed.**

Thanks for your question. After the data collected by the LCS, the raw data should be preprocessed. The PM3006 particulate matter sensor can output six kinds of particle range (i.e., >0.3μm, >0.5μm, >1.0μm, >2.5μm, >5.0μm and >10μm, respectively). By subtracting the six particle range values in turn, the individual particle counters are obtained, and expressed as $x_{0.5}$, $x_{1.0}$, $x_{2.5}$, $x_{5.0}$ and $x_{10.0}$, listed in Table 1, the measured particle number concentration is converted to PM mass concentrations in the $PM_{2.5}$ and $PM_{10}$ size fractions.

The particle counter terms are pretreated and individual from each other. The multi-input one-response preprocessing and prediction models can be written as Eq. (1) to obtain the concentrations $Y_{pm2.5}$.

$$Y_{pm2.5}=w_{1\_pm2.5} \cdot x_{0.5} + w_{2\_pm2.5} \cdot x_{1.0} + w_{3\_pm2.5} \cdot x_{2.5}+w_{4\_pm2.5} \cdot T+w_{5\_pm2.5} \cdot RH + b_{pm2.5}, \quad (1)$$

Where $W_{pm2.5}= [w_{1\_pm2.5}, w_{2\_pm2.5}, w_{3\_pm2.5}, w_{4\_pm2.5}, w_{5\_pm2.5}]$ is the corresponding weight coefficients; the $X_{pm2.5} = [x_{0.5}, x_{1.0}, x_{2.5}, T, RH]$ represents the individual particle counters, the temperature sensor and humidity sensor; the $b_{pm2.5}$ is the intercept values of the model.

To obtain the concentration $Y_{pm10}$, the multi-input one-response preprocessing and prediction models can be written as Eq. (2).

$$Y_{pm10}=w_{1\_pm10} \cdot x_{0.5} + w_{2\_pm10} \cdot x_{1.0} +$$
$$w_{3\_pm10} \cdot x_{2.5}+w_{4\_pm10} \cdot x_{5.0}+w_{5\_pm10} \cdot x_{10.0}+w_{6\_pm10} \cdot T+w_{7\_pm10} \cdot RH + b_{pm10}, \quad (2)$$

Where $W_{pm10}= [w_{1\_pm10}, w_{2\_pm10}, w_{3\_pm10}, w_{4\_pm10}, w_{5\_pm10}, w_{6\_pm10}, w_{7\_pm10}]$ is the corresponding weight coefficients; the $X_{pm10} = [x_{0.5}, x_{1.0}, x_{2.5}, x_{5.0}, x_{10.0}, T, RH]$ represents the individual particle counters, the temperature sensor and humidity sensor; the $b_{pm10}$ is the intercept values of the model.

**(5) Minor comments:**
**p1,l10: a typo "algorithms"**
**p2,l5: additional parentheses**
**Table 4: a typo II, III for O3**
Thanks for your suggestion. The errors are revised in the new vision.

---

## Author Comment (AC3)

**RC2:**

**This study presents a low-cost multi-parameter air quality monitoring system (LCS) that incorporates diverse machine learning algorithms. While the utilization of GA-BP techniques is emphasized, the paper falls short in clearly elucidating its novelty, thereby preventing a comprehensive understanding of its unique contribution. Additionally, the presence of several structural weaknesses within the paper necessitates significant revisions to enhance its coherence and overall quality.**

**Specific Comments:**

**Introduction**

**(1)   The introduction would benefit from a more comprehensive review of recent literature. Additionally, the presence of unclear terms, such as "multi-dimensional multi-response" on line 16, requires clarification to ensure a precise and unambiguous understanding. In addition, the rationale behind the selection of the five specific algorithms used in the study remains unclear. Providing a clear justification for the choice of these algorithms would enhance the understanding of the research methodology and its relevance to the study's objectives.**

Thanks for your question. The "multi-dimensional multi-response" means "multi-input multi-output (MIMO)", and is revised in the new version. The new introduction is revised in the new version.

[revised manuscript text omitted]
 would greatly benefit from expansion to ensure a comprehensive understanding of the study. Specifically, there is a need for more clarity regarding the data collection process, including details on the quantity of data collected for each pollutant and any procedures employed for outlier removal.**
Thanks for question.

The time taken for one set of data collection was one minute and repeated 4 times. The outlier data of the 4 sets of data was eliminated by using the Dixon principle. The remained data was used to get the mean values for each experiment. The values of the LCS and reference instruments were separately logged to the server with the interval of 5 minutes.

**(3) Additionally, in Section 2.1, the inclusion of a map illustrating the data collection site would provide crucial contextual information.**

[Figure]

Figure 1 **Location of the air quality monitoring station during the measurement period**
Thanks for question.

Measurements for gas-phase pollutants and particle pollutants were made continuously between 1 March 2021 and 28 February 2022, which were used as the start and end dates for the analyses. The location, shown in Figure 1, was 30 Yaochang Streat, Zhongyuan District, Zhengzhou City, Henan Province.

**(4) In Section 2.2, it is essential to specify the precise names of the sensors used or provide access to datasheets, particularly for the Alphasense sensors. Clarifying whether the PM sensor used is named PM300S, for instance, would enhance the transparency of the study.**

Thanks for your question. The LCS uses the commercially available particulate matter sensor (PM3006, Cubic sensor and Instrument Co., China) and electrochemical $SO_2$, $NO_2$, $O_3$, CO sensors (B4, Alphasense, UK), respectively.

**(5) Moreover, the paragraph discussing laboratory tests requires expansion. Given the apparent linearity of the sensor response to concentrations, it is necessary to explicate the rationale behind testing non-linear methods. Exploring concentration curves at various temperatures and humidity levels would contribute to a more thorough analysis.**

Thanks for your question. The rationale behind testing non-linear methods is as follows.

1) The linear or multivariate linear calibration models have been developed. However the performance is poor on ambient data, because the output voltages of the four type of gaseous sensors were nonlinearly fluctuated with the linearly increasing temperature and the relative humidity (RH) (Cui et al., 2021).

2) The cross-interference between the four types of gaseous sensors is another problem that needs to be addressed. The cross-interference effect is nonlinear.

The laboratory test is conducted with the purpose to check whether the sensors work effectively before installed into the LCS, calibrated with the different models and used in real-world conditions. The linearity of the gas sensors is tested under steadily increased concentration, which is from 0 - 5 mgm$^{-3}$ for CO sensor, 0 - 0.2 mgm$^{-3}$ for $NO_2$, 0 - 1.1 mgm$^{-3}$ for $O_3$ and 0 - 1.4 mgm$^{-3}$ for $SO_2$ with five more test points, shown in 错误!未找到引用源。. Since the units of outputs of the reference instruments and the sensors were different, the slope was not expected to be 1(Cui et al., 2021). From the 错误!未找到引用源。, we can tell that the $R^2$ for the gas sensors were more than 0.93, which indicated that these gas sensors had good linear responses before calibration, and verified the sensor working properly and effectively and could be applied to the LCS.

**(6) Lastly, directly citing the manufacturer of the reference monitors mentioned on line 23 of page 5, as well as providing information on the methodology employed for the weekly calibrations, would significantly strengthen the study's transparency.**

Thanks for your question.

According to the technical specifications for operation and quality control of ambient air quality continuous automated monitoring system for $SO_2$, $NO_2$, $O_3$ and CO of China(China, 2018), and the technical guide for automatic monitoring by beta ray method for particulate matter in ambient air ($PM_{10}$ and $PM_{2.5}$) (China, 2020), the reference gas and particulate analyzers are checked and calibrated weekly and monthly, respectively.

(7) Calibration method

**The equation (2) seems unclear; there might be a typographical error with X' instead of X.**

Thanks for your question. The error is revised in the new version.

**(8) Furthermore, in section 3.2, it would be beneficial to include additional statistics such as Mean Squared Error (MSE) or Mean Absolute Error (MAE) to provide a more comprehensive evaluation of the model's performance.**

Thanks for your question. The revised version is as follows.

To quantitatively compare the performance of the five calibration models applied to the LCS, and balance the disadvantages of the different metrics, the determination coefficient ($R^2$), root mean square error (RMSE) (Janabi et al., 2021), mean square error (MSE) and mean absolute error (MAE) are utilized. The $R^2$ reflects the fit degree between the model output data and the reference monitor measurement. The measurement results should meet the requirements of environmental standards of China(Jiao et al., 2016). The RMSE measures how much error there is between the predicted values and the reference measurements, and is sensitive to extreme values(Chai & Draxler, 2014). The MAE is a disadvantage against RMSE and a good choice to evaluate the error when the distribution is not Gaussian (Reza, Behzad, & Gulen, 2023). The formula for each of the evaluation metrics are presented as equations (5)-(8), respectively.

$$R^2 = 1 - [\, \textstyle\sum_{i=1}^{n} (y_i - \hat{y}_i)^2 \,] / [\, \textstyle\sum_{i=1}^{n} (y_i - \bar{y}_i)^2 \,], \tag{5}$$

$$\text{RMSE} = \sqrt{\frac{1}{n}\textstyle\sum_{i=1}^{n} (y_i - \hat{y}_i)^2}, \tag{6}$$

$$\text{MSE} = \frac{1}{n}\textstyle\sum_{i=1}^{n} (y_i - \hat{y}_i)^2, \tag{7}$$

$$\text{MAE} = \frac{1}{n}\textstyle\sum_{i=1}^{n} |y_i - \hat{y}_i| \tag{8}$$

Where $\hat{y}_i$, $y_i$ and $\bar{y}$ represent the $i$th model output data form the algorithm-based LCS system, the reference data from the reference instrumentations, and the mean value of the reference instrumentations, respectively. The $n$ is the number of the measurement data in the dataset.

**(9) Results and discussion**
**Paragraph 4.1 presents intriguing insights; however, it could benefit from a clearer presentation. For instance, the method of determining the number of trees in the random forest is not explicitly elucidated.**

Thanks for your question. The revised version is as follows.

For the RF model, the number of trees is determined by using grid search method, which will search the optimal hyper-parameter by traversing a given hyper-parameter combination (Zhu, Zhu, Zhou, Zhu, & Zhang, 2022). A total of 11 kinds of tree numbers are set between 2 and 22. By using grid search to traverse these 11 kinds of tree numbers to obtain different $R^2$.

**(10) Additionally, while it is evident that a sub-period was chosen for testing, the rationale behind this selection remains unexplained. Clarifying these aspects would enhance the overall coherence and understanding of the paragraph.**

Thanks for your question. For instance, the $R^2$ for different gas pollutants within 1 March 2021 and 30 June 2021 are shown in Figure 7. The $R^2$ is improved as the number of trees increasing. The rate of increase and the variation of $R^2$ is negligible beyond 20. The terminal node is specified using a maximum number of sub-node points per node. The $R^2$ is also improved as the number of sub-nodes increasing under the same tree number. The rate of increase and the variation of $R^2$ is negligible

beyond 100. More number of the tree or the sub-node incur higher computational cost and time for the training and small performance improvement. Using this method, the same number of trees can be obtained with the different gas pollutants within 1 July 2021 and 31 October 2021, 1 November 2021 and 28 February 2022.

**(11) In paragraph 4.2, including the size of each segment, as well as the reference temperature, humidity, and concentration range, would enhance the comprehensiveness of the experimental setup and contribute to a more detailed understanding of the study.**

Thanks for your question. The revised version is as follows. The temperature and humidity results are also provided.

[Figure]

t

**Figure 9 Temperature/relative humidity ranges during the measurement period**

During the measurement period, the ranges of the ambient temperature and relative humidity separately were -5°C to 50°C and 10% to 98%., shown in Figure 9. The ambient temperature increased, decreased and fluctuated separately within 1 March 2021 and 30 June 2021, 1 July 2021 and 31 October 2021, 1 November 2021 and 28 February 2022, dividing the whole measurement period into three segments.

**(12) In Figure 9 (b), including a normalized version of the Root Mean Square Error (RMSE) would be beneficial to enable an accurate comparison among the three periods. The same principle applies to Table 3; including a normalized version of the Root Mean Square Error (RMSE) would facilitate an accurate comparison among the different parameters.**

Thanks for your question. The RMSE in table is obtained using equation (6). Where $\hat{y}_i$ and $y_i$ represent the $i$th model output data form the algorithm-based LCS system, and the reference data from the reference instrumentations, respectively. Thus, the result is the normalized version. The wrong expression is revised in the new version.

$$\mathrm{RMSE} = \sqrt{\frac{1}{n}\sum_{i=1}^{n}(y_i - \hat{y}_i)^2}, \tag{6}$$

**(13) Furthermore, the text mentions a division into train and test sets. It would be valuable to clarify whether a cross-validation was also conducted to ensure the robustness of the model evaluation.**

Thanks for your question. To ensure the robustness of the model evaluation, the 5-fold cross validation is also conducted. The dataset is divided into 5 mutually exclusive subsets with same size, the 4 subset is randomly selected as the training set each time, and the remaining 1 subset is used as the test set. After completing each round of validation, the 4 copies are selected again to train the model and the remaining 1 copy is used for validation. After several rounds (less than 5), the loss function is selected to evaluate the optimal model and parameters (Mahesh et al., 2023; Zimmerman et al., 2018).

**(14) These consideration are also valid for the results of gas measurements. Moreover it would be insightful to include a discussion of the varying results obtained for each segment. For instance, a better detailed analysis of why the performance of SO2 is consistently good for period II but considerably poorer for the other periods would enrich the understanding of the data and provide valuable insights into the underlying factors influencing the results.**
Thanks for your question. The revised version is as follows.
For the data of $SO_2$, the results of RF are better than the ones of other methods, and have little difference among the three periods. However, the performances of other methods (MLR, KNN, BP, GA-BP) are poorer for the first and third periods. There may be some reasons for this phenomenon. The first one is the cross interference effect from $NO_2$ and $O_3$, which have the wide range of fluctuations (from about 20 $\mu gm^{-3}$ to 125 $\mu gm^{-3}$) and increasing tendency in period I, respectively. The $NO_2$ and $SO_2$ can react chemically under certain conditions to produce sulfuric acid ($H_2SO_4$) and nitric acid ($HNO_3$), which will affect the reading of $SO_2$ sensor. The $O_3$, highly oxidizing gas, may react with $SO_2$ to form $H_2SO_4$ or sulfite ($H_2SO_3$), resulting in inaccurate sensor readings. The second one is the ambient temperature has a wide range of fluctuations (from about minus 5 °C to plus 45 °C) during the first and third periods, which will affect the stability of electrode material and the readings of the sensor. The last one is the concentration of ambient $SO_2$ is high (more than 30 $\mu gm^{-3}$) in period I and period III, beyond the actual measurement range of the $SO_2$ sensor, which will be researched in future.

**(15) Finally, there is a typo on line 6 of page 14 (dada instead of data).**
Thanks for your question. The typo is revised in the new version with the red color.

(16) Conclusion
**The conclusion paragraph would benefit from a more explicit discussion on the presence of a recommended algorithm for calibration and a thorough examination of its potential limitations. By addressing the challenges associated with generalizing black box models, notably random forests, the conclusion could provide a more nuanced understanding of the practical implications and constraints that may arise from the study's findings.**
Thanks for your question. the conclusion is revised in the new verison.
A low-cost air quality monitoring system (LCS) based on RF, MLR, KNN, BP, GA-BP algorithms were proposed. The system can measure gas-phase pollutants ($SO_2$, $NO_2$, CO and $O_3$) and particle pollutants ($PM_{2.5}$ and $PM_{10}$), simultaneously. With the purpose to estimate the performance of the five algorithms, the LCS was mounted at the same location (Zhengzhou City, China) and consistent height with the reference monitoring system. The measurement was made continuously from 1 March 2021 to 28 February 2022, with the ranges of the ambient temperature and relative humidity

separately minus 5℃ to plus 50℃ and 10% to 98%. The values of the LCS and reference instruments were separately logged to the server for further comparative analysis.

With the pretreated and individual particle counters, $T$ and $RH$ as input, and the concentrations of $PM_{2.5}$ and $PM_{10}$ measured by the reference instrumentation separately as output, the multi-input one-output evaluation models based on RF, MLR, KNN, BP, GA-BP algorithms can be obtained. With the four types of electro-chemical sensors raw data, $T$ and $RH$ as input, and the measurements from the reference monitors as output, the multi-input multi-output evaluation models based on the five algorithms can be obtained. The performance of the calibration models are quantitatively compared by utilizing $R^2$, RMSE, MSE and MAE.

The experimental results show that the $R^2$ of RF for the PM is better than 0.98; the $R^2$ of MLR for the PM is less than 0.91; the $R^2$ of the other three model are within 0.86 and 0.98. The $R^2$ of RF for the gas pollutants ($SO_2$, $NO_2$, CO and $O_3$) is better than 0.93; the $R^2$ of KNN, BP and GA-BP for the gas pollutants ($SO_2$, $NO_2$, CO and $O_3$) is within 0.27 to 0.97; the $R^2$ of MLR for the $NO_2$, CO and $O_3$ is within 0.46 to 0.90, but for SO2 less than 0.40, and even less than 0.1.

The maximum RMSE values of $PM_{2.5}$, $PM_{10}$, $O_3$, CO, $NO_2$, and SO2 between the reference data and the RF, MLR, KNN, BP, GA-BP-based algorithms data are 5.49, 18.68, 13.05, 14.35, and 14.35; 10.37, 45.05, 27.08, 23.10, and 23.65; 4.08, 17.79, 10.57, 14.67, and 14.40; 0.06, 0.23, 0.16, 0.18, and 0.18; 3.99, 14.54, 9.61, 11.07, and 11.21; 2.84, 28.80, 16.44, 21.39, and 21.16, respectively. The maximum MAE values of $PM_{2.5}$, $PM_{10}$, $O_3$, CO, $NO_2$, and SO2 between the reference data and the RF, MLR, KNN, BP, GA-BP-based algorithms data are 3.45, 12.80, 8.31, 9.55, and 9.54; 5.28, 23.20, 13.35, 15.26, and 15.43; 2.88, 13.46, 7.33, 11.14, and 10.90; 0.05, 0.19, 0.11, 0.14, and 0.14; 2.80, 11.08, 6.85, 8.27, and 8.41; 1.16, 4.24, 2.84, 3.43, and 3.40, respectively.

It is should be noted that the results of RF are better than the ones of other methods, have very good agreement with the reference monitors, and have little difference among the three periods. However, the performances of other methods (MLR, KNN, BP, GA-BP) have poor agreement, especially during the first and third periods, which are the winter and spring. There may be some reasons, such as the cross interference effect, the wide range of fluctuation of the climatic factors, and the limitation of the actual measurement range and precision.

Overall, we conclude that, with careful data management and calibration using the machine learning algorithms, especially the RF method, these measurements are consistent with the national environmental protection standard requirement of China, the LCS may significantly improve our ability to spatial heterogeneity in air pollutant concentrations. The air pollutant maps will assist researchers, policymakers, and communities in developing new policies or mitigation strategies to enhance human health. In the next research, we will focus on improving the matching of the measurement precise and range, the generalization of the algorithms in more applications, and the performance of the $SO_2$ sensor.

---

## Author Response (AR2)

**1. RMSE and MSE are the same indicator, so it is no need to report both. Since RMSE and MAE are comparable, MSE should have removed.**

Thanks for your suggestion. The MSE is removed in the new version.